# Total Metabolic Tumor Volume on 18F-FDG PET/CT Is a Useful Prognostic Biomarker for Patients with Extensive Small-Cell Lung Cancer Undergoing First-Line Chemo-Immunotherapy

**DOI:** 10.3390/cancers15082223

**Published:** 2023-04-10

**Authors:** Julia Grambow-Velilla, Romain-David Seban, Kader Chouahnia, Jean-Baptiste Assié, Laurence Champion, Nicolas Girard, Gerald Bonardel, Lise Matton, Michael Soussan, Christos Chouaïd, Boris Duchemann

**Affiliations:** 1Department of Nuclear Medicine, AP-HP, Avicenne University Hospital, 93000 Bobigny, France; 2Department of Nuclear Medicine, AP-HP, European Hospital Georges-Pompidou, University of Paris, 75015 Paris, France; 3Department of Nuclear Medicine, Institut Curie, 92210 Saint-Cloud, France; 4Laboratoire d’Imagerie Translationnelle en Oncologie, Inserm, Institut Curie, 91401 Orsay, France; 5Department of Medical Thoracic and Medical Oncology, AP-HP, Avicenne University Hospital, 93000 Bobigny, France; 6Unité de Pneumologie, CHU Henri Mondor, UPEC, 94000 Créteil, France; 7Institut du Thorax Curie Montsouris, Institut Curie, 75005 Paris, France; 8Paris Saclay, UVSQ, UFR Simone Veil, 78180 Versailles, France; 9Nuclear Medicine, Centre Cardiologique du Nord, 93200 Saint-Denis, France; 10Department of Pneumology, Centre Hospitalier Inter-Communal de Créteil, Paris-Est University, 94010 Créteil, France; 11Inserm UMR 1272 “Hypoxie et Poumon”, UFR SMBH Léonard de Vinci, Université Sorbonne Paris Nord, 93000 Bobigny, France

**Keywords:** small-cell lung cancer, immunotherapy, prognostic, predictive, biomarker, 18F-FDG PET/CT

## Abstract

**Simple Summary:**

Chemotherapy with immune checkpoint inhibitors is the new standard of care for first-line systemic therapy in extensive small-cell lung cancer. The identification of biomarkers for patients that are likely or not likely to respond to such therapy is critical. We aimed at determining whether imaging could help to predict outcomes among these patients. We showed that the total metabolic tumor burden, extracted from pre-treatment 18-FDG PET/CT imaging, may be a useful biomarker associated with survival. Finally, we think that this result should be taken into account in clinical trials, and that it might need further validation through large, independent, and prospective cohorts.

**Abstract:**

*Background*: We aimed to evaluate the prognostic value of imaging biomarkers on 18F-FDG PET/CT in extensive-stage small-cell lung cancer (ES-SCLC) patients undergoing first-line chemo-immunotherapy. *Methods:* In this multicenter and retrospective study, we considered two cohorts, depending on the type of first-line therapy: chemo-immunotherapy (CIT) versus chemotherapy alone (CT). All patients underwent baseline 18-FDG PET/CT before therapy between June 2016 and September 2021. We evaluated clinical, biological, and PET parameters, and used cutoffs from previously published studies or predictiveness curves to assess the association with progression-free survival (PFS) or overall survival (OS) with Cox prediction models. *Results:* Sixty-eight patients were included (CIT: CT) (36: 32 patients). The median PFS was 5.9:6.5 months, while the median OS was 12.1:9.8 months. dNLR (the derived neutrophils/(leucocytes-neutrophils) ratio) was an independent predictor of short PFS and OS in the two cohorts (*p* < 0.05). High total metabolic tumor volume (TMTV^high^ if > 241 cm^3^) correlated with outcomes, but only in the CIT cohort (PFS for TMTV^high^ in multivariable analysis: HR 2.5; 95%CI 1.1–5.9). *Conclusion:* Baseline 18F-FDG PET/CT using TMTV could help to predict worse outcomes for ES-SCLC patients undergoing first-line CIT. This suggests that baseline TMTV may be used to identify patients that are unlikely to benefit from CIT.

## 1. Introduction

At diagnosis, on average, two-thirds of patients with small-cell lung cancer (SCLC) have extensive-stage (ES) disease. The prognosis is very poor, with a 5-year survival rate of 7% [1]. Despite initial high response rates with chemotherapy, the majority of patients relapse rapidly [2]. Recently, immune checkpoint inhibitors (ICIs) have demonstrated improved survival and antitumoral responses in patients with ES-SCLC. Consequently, atezolizumab or durvalumab, both program-death ligand 1 (PD-L1) inhibitors, in association with platinum-etoposide chemotherapy, became the new standard of care for first-line ES-SCLC patients [3,4,5].

Only a few ES-SCLC patients experience a long-term benefit under ICI, and it is much more difficult to predict this benefit at an individual level in comparison with metastatic non-small cell lung cancer patients. To date, there are no useful predictive biomarkers to select the subgroup of patients that might benefit from immunotherapy in the long term. Therefore, the identification of biomarkers for patients that are likely to respond to ICI therapy is critical, and might be a relevant step in enhancing the standard of care for ES-SCLC patients [6].

In the era of immunotherapy and personalized medicine,18F-FDG PET/CT is a functional molecular imaging modality used to assess several aspects of the disease status of oncological patients, including disease staging, prognosis, and response to therapy [7]. Among the metabolic parameters extracted from 18F-FDG PET/CT, the total metabolic tumor volume (TMTV), reflecting the tumor burden (TB), has been shown to be a prognostic factor for overall survival (OS) and progression-free survival (PFS) in patients treated with chemotherapy in several metastatic malignancies, especially in lung cancer [8,9].

Recent evidence has also suggested that a high TB harms anticancer immunity; hence, a high metabolic tumor volume in patients treated with ICI was also shown as a significant prognostic biomarker in some cancers [6,10,11,12]. For SCLC, the most important prognostic parameter is the disease stage, and to a lesser extent age, sex, performance status, lactate dehydrogenase (LDH), and albumin as well [13].

However, to our knowledge, the prognostic value of baseline metabolic tumor burden on 18F-FDG PET/CT has never been evaluated yet in patients with ES-SCLC treated using chemo-immunotherapy.

The present study aimed to determine if 18F-FDG PET/CT parameters can predict outcomes in patients with ES-SCLC undergoing first-line CIT. To assess the possible prognostic or predictive impact on the outcomes, we also included a control group of patients treated using chemotherapy alone.

## 2. Materials and Methods

### 2.1. Patients

We carried out a retrospective study across multiple centers of patients with ES-SCLC who were treated with either chemotherapy alone or a combination of chemotherapy and first-line immune checkpoint inhibitors (ICIs). These patients underwent 18-FDG PET/CT scans at baseline, between June 2016 and September 2021, as part of standard care in three French centers. From May 2019, patients could receive immunotherapy from an early-access program in ES-SCLC. For the CT group, we analyzed a historical cohort of consecutive patients treated before this early-access program.

The flow-chart is provided in Figure 1. The study was approved by our local institutional review board (CLEA-2022-239).

Patients were eligible if they had: (i) biopsy-proven SCLC, (ii) stage IV disease, or (iii) treatment with platinum-etoposide chemotherapy alone (CT group) or in association with first-line PD-L1 inhibitors, atezolizumab, or durvalumab (CIT group). Patients were excluded if: (i) the delay between FDG PET/CT and the first ICI perfusion was >7 weeks (n = 7), (ii) the follow-up was <6 months, defined as the delay between the last contact and the diagnostic (n = 7), or (iii) they had another primary malignancy (n = 10). The inclusion criteria for blood samples assessment before ICI and chemotherapy was 7 days. Demographic, clinical, pathological, biological, and molecular data were also collected.

### 2.2. FDG PET/CT Acquisitions

After a fasting time of at least 6 h, 18F-FDG PET/CT scans were performed 60 min (median 60 min; range 44–73) after the injection of 18F-FDG (median activity 219 MBq; range: 99–411). In most cases, images were obtained from the skull vertex to the proximal femur. Images were acquired and reconstructed according to current guidelines [14] using six PET/CT scanners (Gemini TF Philips Medical system: n = 9 patients, General Electric Discovery MI: n = 32 patients, General Electric Discovery 710: n = 18 patients, General Electric Discovery IQ: n = 4 patients, Philips Vereos: n = 5 patients). Finally, images were interpreted by two experienced physicians, board-certified in nuclear medicine (RDS and JGV).

### 2.3. Measurement of Biological and Imaging Biomarkers

All hypermetabolic metastatic lesions were selected for the analysis, while hypermetabolic foci explained by inflammatory or physiologic activity were excluded. As a measure of the tumor glycolytic activity, the 18F-FDG PET/CT uptake was quantified by the maximum standardized uptake values normalized by body weight (SUVmax). To assess the metabolic tumor burden, MTV was measured by setting a margin threshold of 42% of SUVmax [15]. The tumor SUVmax was the maximum SUV of all of the lesions in a patient. TMTV was defined as the sum of the individual MTVs of all of the lesions analyzed. Blood cell counts at baseline before ICI treatment (within 7 days before the first treatment) were extracted from electronic medical records.

### 2.4. Outcomes: Progression-Free Survival and Overall Survival

The primary endpoint analysis was to build a multivariable model using pretreatment 18F-FDG PET imaging to predict the outcome. Progression-free survival (PFS) was defined as the delay between the first CIT or the CT perfusion, and the disease progression or the death from any cause; and overall survival (OS) was defined as the delay between the first ICI-chemotherapy or chemotherapy administration, and the death from any cause or censoring at the time when the patient was known to be alive. Follow-up was determined from the first ICI-chemotherapy or chemotherapy perfusion to the date of the last clinical consultation. The assessment of outcome was blinded.

### 2.5. Statistical Analysis

For the derived neutrophils/(leucocytes–neutrophils) ratio (dNLR), we used the cutoff of 3 (>3 vs. ≤3) from the largest published study with ICIs [16]. For biomarkers extracted from 18F-FDG PET imaging (TMTV and Tumor SUVmax), we used predictiveness curves to determine the relevant cut-off values [17,18] using 6-month PFS as a state variable based on the results of the pivotal IMpower133 study [3]. ROC curves were obtained to determine AUC for the prediction of survival outcomes at several time horizons. To analyze the connections between different factors, Spearman’s rank correlation coefficients were computed. Survival curves were estimated for each group using the Kaplan-Meier method, and compared using the log-rank test. The potentials of all pretreatment blood and [18F]-FDG PET biomarkers for predicting survival were examined using Cox models. The Cox proportional hazard regression models were used in multivariable analyses after backward variable selection, to identify significant independent factors. The likelihood ratio test (LRT) for the added prognostic values of TMTV and dNLR were obtained by comparing the log likelihoods of multivariable prognostic models without TMTV or dNLR, and with TMTV plus dNLR (chi-square test: χ2). All statistical tests were two-sided, and a *p*-value of less than 0.05 was deemed statistically significant. Analyses were performed with RStudio (version 1.3.1073, 2009–2020 RStudio, PBC).

## 3. Results

### 3.1. Patient Characteristics

Table 1 summarizes the initial characteristics of the patients. We analyzed 68 patients with ES-SCLC, 36 from the CIT group, and 32 from the CT group. The median age was 67 years old (with a range of 50 to 84 years old), and 50 of the patients (which represents 74% of the total) were male. Patients had a median of two metastatic sites (range, 1–8), 64 patients (94%) had a smoking history, 26 patients (38%) had liver metastasis (LM+), and 22 (32%) had brain metastasis. The median follow-up for the entire population was 11.9 (95%CI 8.5–16.3) months, 12.2 (95%CI 9.6–17.0) months in the CIT group, and 9.8 (95%CI 7.5–15.3) in the CT group. Thresholds determined by predictiveness curves were as follows: TMTV > 241 cm^3^ vs. ≤241 and Tumor SUVmax > 12 vs. ≤ 12 (Appendix A).

### 3.2. Correlation between Biomarkers

Correlations between PET biomarkers, extracted from tumor lesions and variables of interest, are presented in Appendix A. We found that biomarkers extracted from 18F-FDG PET/CT imaging (TMTV and tumor SUVmax) did not correlate significantly with each other (*p* > 0.05). Furthermore, there was no significant relationship between these biomarkers, and clinical or biological parameters (Spearman’s correlation coefficient < 0.3). As an example, when testing the relationship between TMTV and dNLR, Spearman’s correlation coefficient was 0.08 (*p* > 0.10).

### 3.3. Univariable and Multivariable Analyses: PFS

The median PFS was 5.9 (95% CI 5.2–7.9) months in the CIT group and 6.5 (95% CI 4.6–7.6) months in the CT group. In the study, progressive disease was observed in 96% of the patients (65 patients), with 34 patients (94%) from the CIT group and 31 (97%) from the CT group. The AUCs for the predictions of PFS at 6, 12, and 18 months in the two groups based on TMTV and dNLR are provided in Appendix A. When analyzing the data using univariable analysis, high dNLR and high TMTV were significantly related to a shorter PFS in the CIT group (Table 2, Figure 2). However, only a high dNLR was correlated with a shorter PFS in the CT group. (Table 3, Appendix A). Moreover, a high tumor SUVmax, liver or brain metastases, age > 70 years, or PS ≥ 2 did not correlate with PFS in the two groups.

After conducting a multivariable analysis, the study found that both a high dNLR and a high TMTV were independent and statistically significant prognostic factors in the CIT group (HR 3.1, 95% CI 1.3–7.2 and HR 2.5, 95% CI 1.1–5.9, respectively). However, only a high dNLR was found to be statistically significant in the CT group (HR 3.3, 95% CI 1.4–7.7).

TMTV and dNLR did not provide any additional prognostic value to the multivariable model obtained with backward elimination for PFS in the CIT group (*p* > 0.05 for both; Appendix A).

### 3.4. Univariable and Multivariable Analyses: OS

The study’s results showed that the median OS was 12.1 months (95% CI 9.6–17.0) in the CIT group, and 9.8 months (95% CI 7.5–15.3) in the CT group. Of the total of 68 patients, 56 patients (which represents 82% of the total) died, with 25 (69%) in the CIT group and 31 (97%) in the CT group. The AUCs for the prediction of OS at 6, 12, and 18 months in the two groups based on TMTV and dNLR are provided in Appendix A.

The univariable analysis revealed that high dNLR and liver metastases (LM+) were both associated with poor OS in the CIT group (Table 2; Figure 3). However, only a high dNLR was found to be associated with a worse OS in the CT group (Table 3, Appendix A). Despite a clear trend in the univariable analysis, a high TMTV was not a statistically significant prognostic factor for OS in the CIT group (HR 2.2, 95%CI 1.0–4.8, *p* = 0.06). High Tumor SUVmax, brain metastases, age > 70 years, and PS ≥ 2 did not correlate with OS in the two groups.

According to the analysis of multiple variables, having a high dNLR was found to be a significant and independent factor for predicting outcomes in both groups (CIT group: HR 2.7, 95% CI 1.1–6.5; CT group: HR 3.2, 95% CI 1.2–8.7). The presence of liver metastases did not remain as a significant predictor for OS in the CIT group (HR 1.7, 95% CI 0.9–5.8).

Furthermore, TMTV and dNLR added significant prognostic values to the multivariable model obtained with the backward elimination process for OS in the CIT group (*p* < 0.05 for both; Appendix A).

### 3.5. Signature Using TMTV and LM Status for the Prediction of Survival

We combined the TMTV and LM status (yes: LM+; no: LM−), both with *p* values < 0.10 in a univariable analysis for PFS and OS in the CIT cohort, and used them to build a signature for baseline risk stratification. Based on these two prognostic factors, we identified three distinct risk groups as low- (TMTV^low^ and LM− n = 18 in the CIT cohort and n = 15 in the CT cohort), intermediate- (TMTV^high^ or LM+ n = 10 in the CIT cohort and n = 10 in the CT cohort), and high risk (TMTV^high^ and LM+ n = 8 in the CIT cohort and n = 7 in the CT cohort).

In the CIT cohort, the median OS was 9.4 months (95%CI 7.9-NA) for the high-risk group versus 13.9 months (95%CI 11.6-NA) for the intermediate risk group versus 14.1 months (95%CI 12.0-NA) for the low-risk group (*p* < 0.01: the statistical difference between the high-risk group versus the intermediate/low-risk groups) (Table 4). At 12 months, the probability of OS was approximately 12% for the high-risk group, versus 51% and 60 % for the intermediate and low risk groups, respectively (Appendix A). This TMTV/LM signature was not associated with PFS in the CIT cohort (Appendix A). In the CT cohort, the comparison of survival probabilities did not produce similar results, since no significant association was detected between our TMTV/LM signature and the outcomes (neither PFS nor OS: Table 4 and Appendix A, Figure 4 and Appendix A).

## 4. Discussion

This study explored the prognostic values of specific metabolic parameters from the pretreatment 18F-FDG PET/CT of ES-SCLC patients undergoing first-line therapy with CIT or CT. The prognosis of patients displaying a high metabolic TB before first-line CIT was worse than patients displaying a low metabolic TB. A similar trend was observed for the OS analysis in the CIT group; however, TMTV did not reach statistical significance.

We identified a relevant cut-off to determine patients displaying a high versus low TMTV (> or ≤241 cm^3^), using predictiveness curves, which is consistent with other studies (a cut-off at 245.7 cm^3^ was used in a recent study for a patient treated with first-line treatment in ES-SCLC [19]). A recent meta-analysis [8] showed that high baseline TMTV is also prognostic for OS and PFS in SCLC patients treated with CT. Our results suggest that TMTV may have a strong prognostic impact in patients treated with CIT. This is evidenced by its significant correlation with PFS in the group treated with CIT. However, it cannot be excluded that TMTV would not have been significantly associated with survival outcomes in patients treated with CT alone if the sample size was larger. Beyond a better understanding of the extent and severity of the disease, further larger studies are warranted to determine whether and how measuring TMTV could be useful for clinicians to tailor their treatment strategies to individual patients and potentially improve their outcomes.

Recent evidence has also suggested that a high TB harms anticancer immunity [6] in several other malignancies, including non-small cell lung cancer (NSCLC) [20]. To our best knowledge, this had never been studied before in ES-SCLC patients treated with CIT. This phenomenon might be explained by a local immunosuppressive effect owing to certain characteristics of the tumor microenvironment. In some instances of high TB, tumor size, levels of inflammation, T cell senescence and exhaustion, and weakening anticancer immunity might explain the lack of response to ICI [6].

In our study, a high tumor SUVmax (>12) was not correlated with OS, either with PFS in ES-SCLC patients treated with CIT, or in patients treated with CT. These results are consistent with previous studies: a recent meta-analysis published in 2021 by Christensen [8], showed that only 7 out of 28 studies showed a significant prognostic value of SUVmax for OS and/or PFS.

Moreover, we showed that liver metastasis was associated with a worse OS for ES-SCLC patients treated with CIT (univariable analysis only), but not for patients treated with CT. Patients with liver metastases often have more aggressive cancers, which also contribute to poorer responses to systemic therapies. Liver involvement is a well-known pejorative indicator that is associated with reduced responses and worse outcomes (PFS and OS) in patients treated with ICI [21]. One reason for this is that the liver is an important site for immune regulation and tolerance, which make it more difficult for the immune system to recognize and to attack cancer cells in this organ [22]. Liver metastases were associated with reduced marginal CD8+ T-cell infiltration and decreased activated CD8+ T cells from the systemic circulation in patients with melanoma and NSCLC [21], and the mouse model [22]: which may explain the diminished immunotherapy efficacy. While the presence of liver metastases suggests a more advanced stage of disease, it may also indicate an increase in difficulty in effectively targeting cancer cells in the liver. Indeed, the liver is a vital organ that plays a critical role in metabolizing drugs, including ICI, which can lead to lower drug concentrations in the liver, and reduced efficacy [23].

A combination of multiple biomarkers could be a relevant way to optimize both prediction and prognostic accuracy [24,25]. We developed a score, combining the TMTV at baseline before therapy and the LM status, stratifying the population into three risk groups. Our research indicates that combining TMTV and LM in a promising signature has a more significant impact on predicting OS. However, this approach needs to be tested and confirmed in larger, independent, and prospective groups.

Multiple previous studies had demonstrated that a high peripheral pro-inflammatory status, which can be measured using biomarkers such as LDH and dNLR, was associated with worse outcomes in patients with cancer [26,27,28], and was correlated with a reduction in the efficacy of immunotherapy [29]. However, several randomized studies comparing Atezolizumab versus Docetaxel in advanced-stage NSCLC suggested these biomarkers had a prognostic rather than a predictive role [30]. The results in our study were consistent with previous findings: a high dNLR was associated with worse outcomes (OS and PFS) in the two groups.

The main limitations of our study are its retrospective nature and its relatively small sample size. We compared patients from a period before and after immunotherapy availability in this setting. Nevertheless, there was no major modification in the standard of care of ES-SCLC for the two cohorts. Furthermore, another strength was the use of predictiveness curves for testing the predictive capacity of candidate biomarkers that enables the most complete analysis. Indeed, predictiveness curves fit into an integrative approach which could give information about risk (as well as logistic regression, Cox models, and Kaplan-Meier analysis) but also classification performance (as well as ROC curves).

## 5. Conclusions

A high TMTV on 18F-FDG PET/CT was correlated with a worse outcome for ES-SCLC patients undergoing first-line CIT. This finding suggests that TMTV on 18F-FDG PET/CT could be used as a prognostic biomarker, identifying patients who shall not benefit from ICIs. Moreover, the TMTV/LM signature could also help to stratify patients at baseline, and guide clinical decision-making for ES-SCLC patients before first-line therapy. Such findings could be taken into account in clinical trials, and they might need further validation through large, independent, and prospective cohorts.

## Figures and Tables

**Figure 1 cancers-15-02223-f001:**
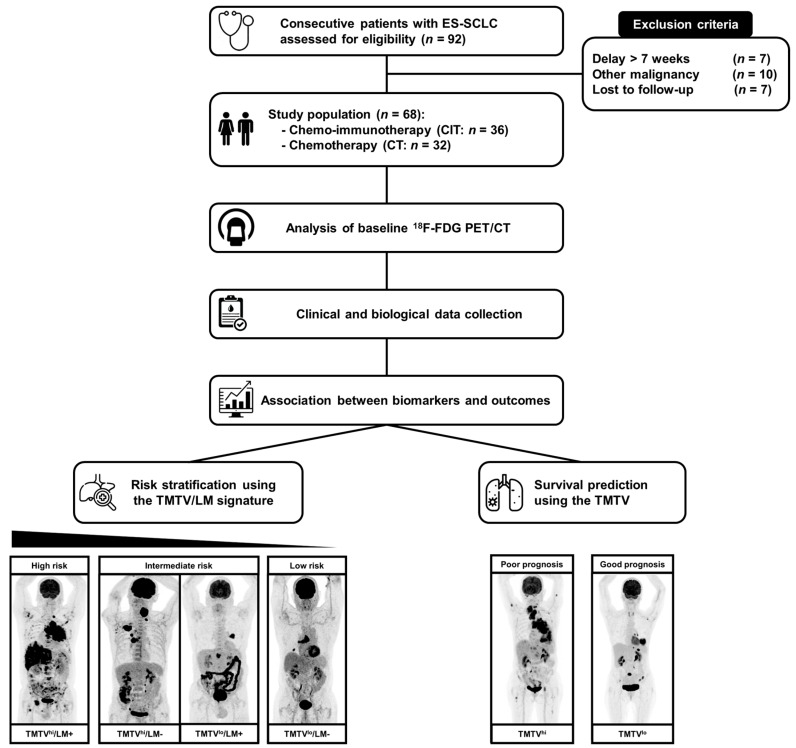
Flow-chart.

**Figure 2 cancers-15-02223-f002:**
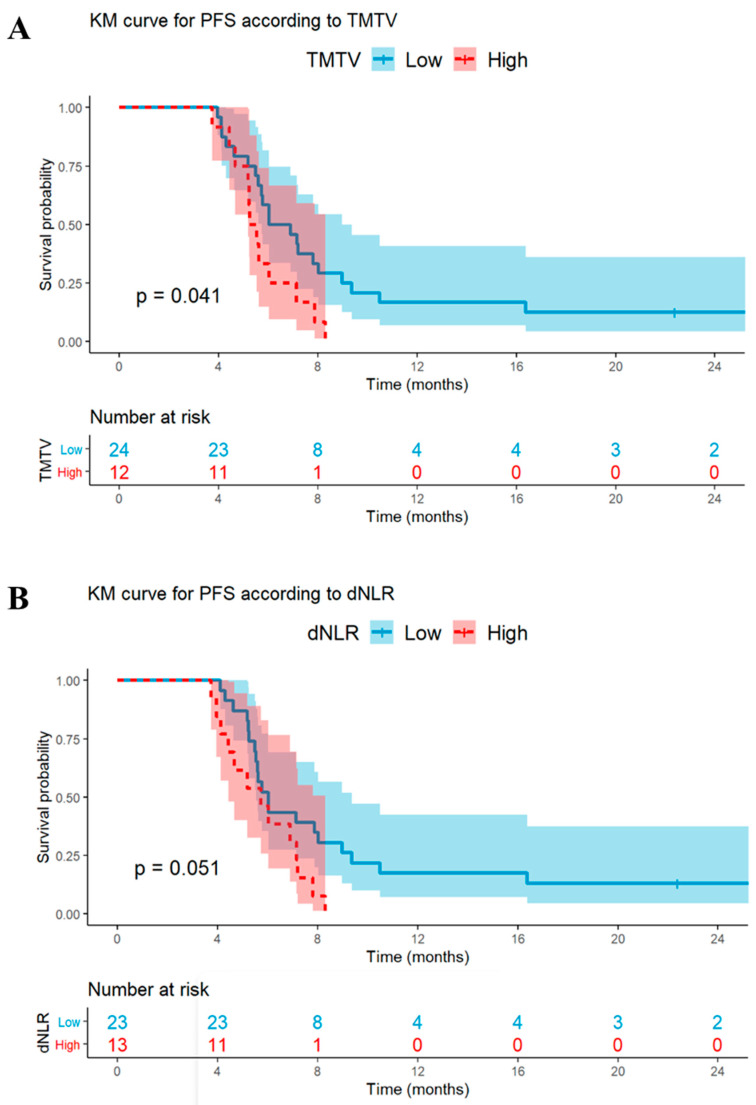
Kaplan-Meier curves of progression-free survival (PFS) according to the Total Metabolic Tumor Volume (TMTV) (**A**) and the derived Neutrophil-to-Lymphocyte Ratio (dNLR) (**B**) in the chemo-immunotherapy (CIT) cohort.

**Figure 3 cancers-15-02223-f003:**
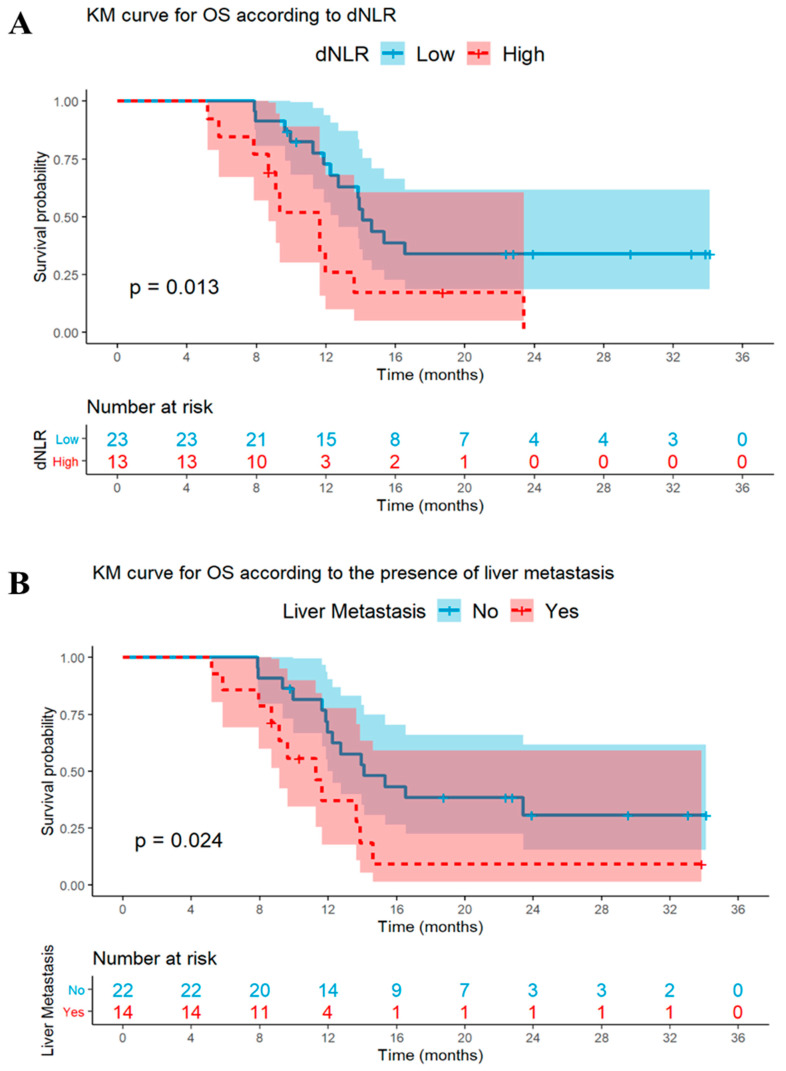
Kaplan-Meier curves of overall survival (OS) according to the derived Neutrophil-to-Lymphocyte Ratio (dNLR) (**A**), and the presence of liver metastases (**B**) in the chemo-immunotherapy (CIT) cohort.

**Figure 4 cancers-15-02223-f004:**
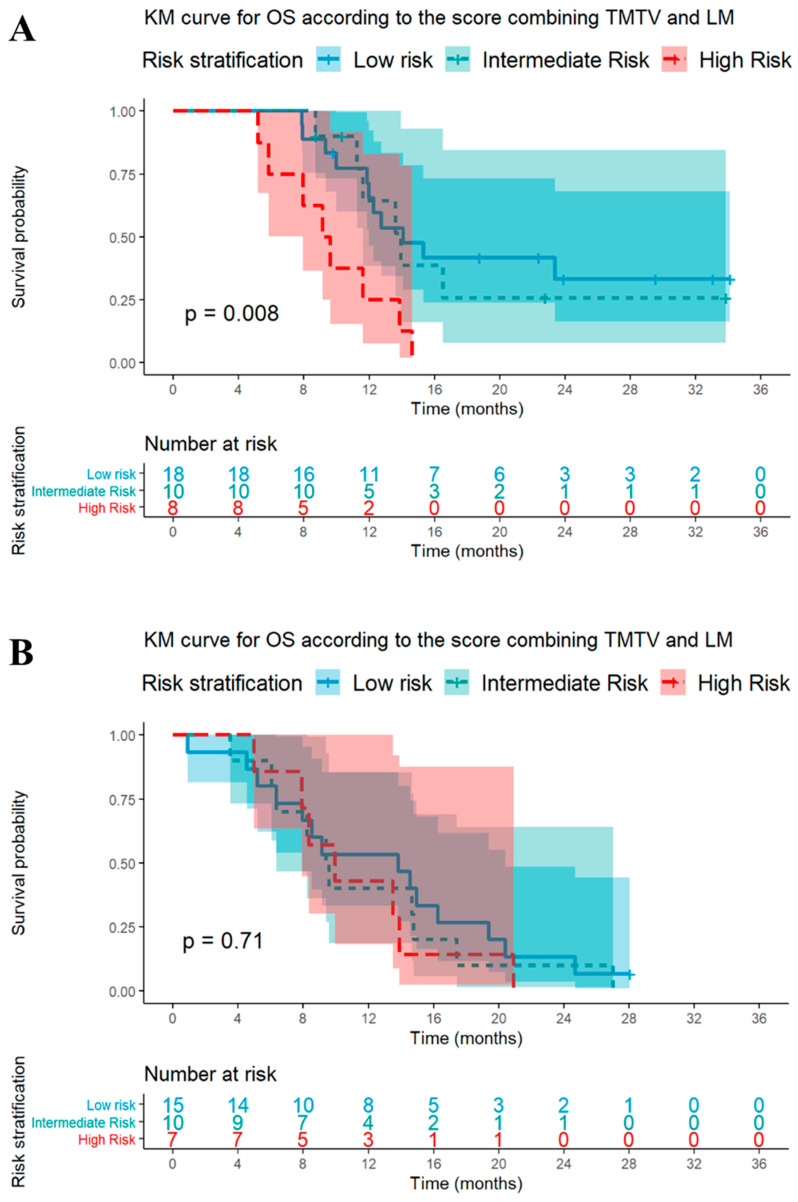
Kaplan-Meier curves of overall survival (OS) based on total metabolic tumor volume (TMTV) and liver metastases (LM) status in the chemo-immunotherapy (CIT) cohort (**A**), and in the chemotherapy (CT) cohort (**B**).

**Table 1 cancers-15-02223-t001:** Patient characteristics.

	All Patients (n = 68)	CIT Group (n = 36)	CT Group (n = 32)
	Median [Range], n (%)	Median [Range], n (%)	Median [Range], n (%)
CLINICAL CHARACTERISTICS			
Demographic parameters			
Age (years)	67 (50–84)	67 (51–84)	66 (50–84)
Gender (Male/Female)	50 (74)/18 (26)	25 (69)/11 (31)	25 (78)/7 (22)
Performance Status (ECOG)	1 (0–3)	1 (0–2)	1 (0–3)
Smoking history (current/former/no/unknown)	31 (46)/33 (49)/3 (4)/1 (1)	13 (36)/20 (55)/2 (6)/1 (3)	18 (56)/13 (41)/1 (3)/0 (0)
Biology			
Leukocytes (G/L)	8.9 (5.8–19.6)	8.7 (6.1–19.6)	9.6 (5.8–16.4)
Neutrophils (G/L)	6.5 (2.0–15.0)	5.9 (2.6–15.0)	6.7 (2.0–12.1)
dNLR	2.5 (0.3–10.1)	2.6 (0.7–9.0)	2.4 (0.3–10.1)
Lymphocytes (G/L)	1.7 (0.3–3.7)	1.7 (0.6–3.0)	1.7 (0.3–3.7)
Staging			
Number of metastatic sites	2 (1–8)	2 (1–8)	2 (1–6)
Liver involvement	26 (38)	14 (39)	12 (37)
Brain involvement	22 (32)	10 (28)	12 (37)
PET IMAGING PARAMETERS			
Tumor glucose uptake			
Tumor SUVmax	12.7 (3.7–44.0)	13.4 (6.0–44.0)	12.3 (3.7–29.3)
Metabolic Tumor Burden			
TMTV	182.9 (1.7–2257.2)	178.5 (3.1–2257.2)	196.3 (1.7–880.6)
TREATMENT			
First-line			
Carboplatin—Etoposide—Atezolizumab	34 (50)	34 (94)	NA
Carboplatin—Etoposide—Durvalumab	2 (3)	2 (6)	NA
Carboplatin—Etoposide	32 (47)	NA	32 (100)
SURVIVAL			
Progression	65 (96)	34 (94)	31 (97)
Death	56 (82)	25 (69)	31 (97)

Abbreviations: chemotherapy (CT), chemo-immunotherapy (CIT), derived neutrophil-to-lymphocyte ratio (dNLR), total metabolic tumor volume (TMTV), maximum standardized uptake value (SUVmax).

**Table 2 cancers-15-02223-t002:** Significance of biomarkers in predicting progression-free survival and overall survival in patients who underwent first-line chemo-immunotherapy with univariable and multivariable analyses.

N = 36 pts	OVERALL SURVIVAL (Events: n = 25)				PROGRESSION-FREE SURVIVAL (Events: n = 34)			
	Univariable		Multivariable		Univariable		Multivariable	
Variable	*p*	HR (95CI)	*p*	HR (95CI)	*p*	HR (95CI)	*p*	HR (95CI)
DEMOGRAPHICS								
Age ≥ 70 years (vs. <70 years)	0.08	2.1 (0.9–4.8)	0.41	1.5 (0.6–3.8)	0.14	1.8 (0.8–3.7)	-	-
Male (vs. female)	0.15	0.5 (0.2–1.2)	-	-	0.45	1.3 (0.4–1.6)	-	-
PS ≥ 2 (vs. <2)	0.63	1.3 (0.4–3.8)	-	-	0.78	1.1 (0.4–2.1)	-	-
BIOLOGY								
High dNLR	0.02	2.7 (1.2–6.0)	0.03	2.7 (1.1–6.5)	0.05	2.1 (0.9–4.3)	<0.01	3.1 (1.3–7.2)
IMAGING								
18F-FDG PET/CT								
High TMTV	0.06	2.2 (1.0–4.8)	0.07	2.3 (0.9–5.8)	0.04	2.2 (1.1–4.6)	0.03	2.5 (1.1–5.9)
High tumor SUVmax	0.75	1.1 (0.4–1.9)	-	-	0.53	1.2 (0.4–1.6)	-	-
Liver metastases (vs. no)	0.03	2.5 (1.1–5.6)	0.26	1.7 (0.9–5.8)	0.57	1.2 (0.6–2.5)	-	-
BRAIN MRI and/or CT								
Brain metastases (vs. no)	0.68	0.8 (0.3–2.0)	-	-	0.48	0.8 (0.4–1.6)	-	-

Notes: Each biomarker’s distribution is originally a continuous variable, but it is converted into a discrete categorization with two categories (high and low) using cutoffs from previously published studies or predictiveness curves: dNLR (>3 vs. ≤3), TMTV (>241 vs. ≤241cm^3^), and tumor SUVmax (>12 vs. ≤12). Abbreviations: hazard ratio (HR), confidence interval (CI), performance status (PS), derived neutrophil-to-lymphocyte ratio (dNLR), total metabolic tumor volume (TMTV), maximum standardized uptake value (SUVmax).

**Table 3 cancers-15-02223-t003:** Prognostic significance of biomarkers for progression-free survival and overall survival in univariable and multivariable analyses for patients treated with first-line chemotherapy.

N = 32 pts	OVERALL SURVIVAL (Events: n = 31)				PROGRESSION-FREE SURVIVAL (Events: n = 31)			
	Univariable		Multivariable		Univariable		Multivariable	
Variable	*p*	HR (95CI)	*p*	HR (95CI)	*p*	HR (95CI)	*p*	HR (95CI)
DEMOGRAPHICS								
Age ≥ 70 years (vs. <70 years)	0.09	1.8 (0.9–3.8)	0.83	1.1 (0.5–2.7)	0.38	1.4 (0.7–3.0)		
Male (vs. female)	0.37	0.7 (0.3–1.6)	-	-	0.10	0.5 (0.2–1.2)	0.58	0.8 (0.3–1.9)
PS ≥ 2 (vs. <2)	0.15	1.9 (0.8–4.7)	-	-	0.90	1.1 (0.4–2.5)	-	-
BIOLOGY								
High dNLR	<0.01	3.4 (1.5–7.9)	0.02	3.2 (1.2–8.7)	0.05	2.2 (1.0–4.8)	<0.01	3.3 (1.4–7.7)
IMAGING								
18F-FDG PET/CT								
High TMTV	0.18	1.7 (0.8–3.6)	-	-	0.28	1.5 (0.7–3.3)	-	-
High tumor SUVmax	0.12	0.6 (0.3–1.2)	-	-	0.68	0.9 (0.4–1.8)	-	-
Liver metastases (vs. no)	0.94	1.0 (0.5–2.1)	-	-	0.28	0.6 (0.3–1.4)	-	-
BRAIN MRI and/or CT								
Brain metastases (vs. no)	0.34	1.4 (0.7–3.0)	-	-	0.29	1.5 (0.7–3.2)	-	-

Notes: The distribution of each PET biomarker is a continuous variable that is transformed into a discrete categorization in 2 categories (high vs. low) using cutoffs from previously published studies or predictiveness curves: dNLR (>3 vs. ≤3), TMTV (>241 vs. ≤241), and tumor SUVmax (>12 vs. ≤12). Abbreviations: hazard ratio (HR), confidence interval (CI), performance status (PS), derived neutrophil-to-lymphocyte ratio (dNLR), total metabolic tumor volume (TMTV), maximum standardized uptake value (SUVmax).

**Table 4 cancers-15-02223-t004:** Estimation of median overall survival (OS) and median progression-free survival (PFS) in the chemotherapy (CIT) cohort, and in the chemotherapy (CT) cohort based on total metabolic tumor volume (TMTV) and liver metastases (LM) status.

Cohort and Strata	Patients (n)	Median OS [95% CI]	Events (n)	*p*-Value	Median PFS [95% CI]	Events (n)	*p*-Value *
CIT cohort	36			<0.01			0.33
High risk	8	9.4 [7.9–NA]	8		5.4 [5.2–NA]	8	
Intermediate risk	10	13.9 [11.6–NA]	6		5.8 [4.7–NA]	10	
Low risk	18	14.1 [12.0–NA]	11		6.6 [5.6–10.5]	16	
CT cohort	32			0.71			0.38
High risk	7	9.9 [7.9–NA]	7		6.5 [5.5–NA]	7	
Intermediate risk	10	9.5 [6.4–NA]	10		7.3 [3.5–NA]	9	
Low risk	15	13.8 [8.0–20.4]	14		6.1 [5.0–8.5]	15	

Abbreviations: confidence interval (CI), not applicable (NA). Legend: * *p* value for the difference between the high-risk group and low/intermediate-risk groups.

## Data Availability

The data presented in this study are available in the article and Appendix A.

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
