# Peer review of "Total Metabolic Tumor Volume on 18F-FDG PET/CT Is a Useful Prognostic Biomarker for Patients with Extensive Small-Cell Lung Cancer Undergoing First-Line Chemo-Immunotherapy"

_cancers, 2023, doi:10.3390/cancers15082223_

Round 1

Reviewer 1 Report

Grambow-Velilla et al present an article studying the prognostic role of tumor volume evaluated by 18F-FDG PET/CT in patients with SCLC treated with chemoimmunotherapy. The paper is well written and the data is presented in a clear and honest fashion. However the article lacks novelty as the findings merely confirm the widely accepted negative prognostic value of high tumor burden. A high TMTV does not reflect a higher glucose avidity by the tumor but a higher number of lesions. Additionally, liver metastases are a well stablished negative predictive factor of response to immunotherapy. 

Author Response

Comment #1: Grambow-Velilla et al present an article studying the prognostic role of tumor volume evaluated by 18F-FDG PET/CT in patients with SCLC treated with chemoimmunotherapy. The paper is well written and the data is presented in a clear and honest fashion.

This is an accurate depiction of our work. We also thank the reviewer for its detailed and insightful comments. Based on the different suggestions, we have made changes in the manuscript and clarified some points concerning our results.

Comment #2: However, the article lacks novelty as the findings merely confirm the widely accepted negative prognostic value of high tumor burden.

We fully agree with this excellent remark. We fully agree with your comment concerning tumor burden, which is a well-established negative prognosticator. However, the link between pre-treatment tumor burden and outcomes has not been studied enough in ES-SCLC patients, especially those who will receive chemo-immunotherapy. The search for specific prognostic biomarkers for ES-SCLC is an urgent priority in the field of lung cancer research, as it has the potential to significantly improve patient outcomes and reduce the burden of this devasting disease. And it is particularly critical for patients undergoing first-line systemic therapy. With this present work, we sought to highlight the clinical utility of several biomarkers extracted from pretreatment [18F]F-FDG PET/CT imaging, including metabolic tumor burden. After further validation, these could help clinicians to tailor their treatment strategies to individual patients and potentially improve their outcomes and reduce the risk of side effects (e.g. combination of chemotherapy and ICI).

Comment #3: A high TMTV does not reflect a higher glucose avidity by the tumor but a higher number of lesions.

Absolutely. But as far as we know, the TMTV measurement takes into account the total volume but also the metabolic activity of all active cancerous lesions in the body, providing a comprehensive assessment of the extent of the disease. Previous literature has shown that TMTV is a robust and valuable tool in assessing the overall disease burden in patients with lung cancer, especially NSCLC. Too few papers have evaluated the prognostic value of pretreatment [18F]F-FDG PET/CT imaging for ES-SCLC. Regarding our results, we think that measuring TMTV could be useful for oncologists to better understand the extent and severity of the disease and also guide treatment decisions. Finally, the potential of TMTV is quite wide since it could also be a serious candidate biomarker for accurate monitoring of response to therapy over time which requires further research.

We have added a sentence in the Discussion accordingly: “Beyond a better understanding of the extent and severity of the disease, further larger studies are warranted to determine whether and how measuring TMTV could be useful for clinicians to tailor their treatment strategies to individual patients and potentially improve their outcomes.”

Comment #4: Additionally, liver metastases are a well-established negative predictive factor of response to immunotherapy.

Thank you for this constructive comment. Response to immunotherapy is a complex and multifactorial process and the presence of liver metastases could be predictive of poor outcomes in a wide range of cancers.  Research has effectively shown that it is a negative prognostic factor for patients receiving immunotherapy. It suggests a more advanced stage of the disease but also an increased difficulty in targeting cancer cells in the liver.

We have thus added several sentences in the Discussion accordingly: “Patients with liver metastases often have more aggressive cancers, which also contribute to poorer response to systemic therapies. Liver involvement is a well-known pejorative indicator that is associated with reduced responses and worse outcomes (PFS and OS) in patients treated with ICI [21]. One reason for this is that the liver is an important site for immune regulation and tolerance, which make it more difficult for the immune system to recognize and attack cancer cells in this organ [22].”

The REF22 is entitled: “Liver metastasis restrains immunotherapy efficacy via macrophage-mediated T cell elimination”, from Yu et al., Nature Medicine, 2021.

“While the presence of liver metastases suggests a more advanced stage of disease, it may also indicate an increased difficulty in effectively targeting cancer cells in the liver. Indeed, the liver is a vital organ that plays a critical role in metabolizing drugs, including ICI, which can lead to lower drug concentrations in the liver and reduced efficacy [23].”

The REF23 is entitled: “Optimizing systemic therapy for advanced hepatocellular carcinoma: the key role of liver function”, from Cabibbo et al, Dig Liver Dis, 2022.

Reviewer 2 Report

Thank you for a well-written and interesting manuscript. Only few comments. 

1. Which factors were included in the multivariable cox-models?

2. Did you perform c-statistics to investigate the AUC before and after addition of high gNLR and high TMTV?

3. The test for median OS in high, intermediate and low risk in the CIT cohort was significicant. Please state more clearly that the test is high vs. low - i doubt that there is a significant difference between 13.9 and 14.1 months. 

Author Response

Comment #1: Thank you for a well-written and interesting manuscript. Only few comments.

In return, we thank the reviewer for its detailed and insightful comments. Based on the different suggestions, we have made changes in the manuscript and clarified some points concerning our results.

Comment #2: Which factors were included in the multivariable cox-models?

Thank you for this excellent remark. We performed a backward variable selection to build Cox proportional hazard regression models and multivariable analyses. We have modified the sentence in the paragraph Statistical Analysis as follows: “The Cox proportional hazard regression models were used in multivariable analyses after backward variable selection to identify significant independent factors.”

In the chemo-immunotherapy group, age, dNLR, TMTV and liver metastasis were included in the multivariable model for OS while only dNLR and TMTV were included in the multivariable model for PFS. In the chemotherapy group, age and dNLR were included in the multivariable model for OS while gender and dNLR were included in the multivariable model for PFS.

Comment #3:   Did you perform c-statistics to investigate the AUC before and after addition of high dNLR and high TMTV?

Thank you very much for this excellent question regarding AUC for dNLR and TMTV.

First, we have added a table in the supplemental material for the calculation of AUC with 95%CI (DeLong test) from ROC curves to predict PFS and OS at three different horizon (6 months, 12 months and 18 months) using TMTV and dNLR in both groups (results presented below in the Table S1).

AUC (95%CI)

PFS

OS

6 months

12 months

18 months

6 months

12 months

18 months

CHEMO-IMMUNOTHERAPY GROUP

dNLR

0.55 (0.35-0.74)

0.70 (0.44-0.95)

0.66 (0.33-1.00)

0.85 (0.61-1.00)

0.72 (0.54-0.90)

0.62 (0.40-0.82)

TMTV

0.59 (0.40-0.79)

0.63 (0.35-0.90)

0.62 (0.45-0.78)

0.88 (0.73-1.00)

0.61 (0.42-0.80)

0.63 (0.44-0.83)

CHEMOTHERAPY GROUP

dNLR

0.61 (0.39-0.82)

0.58 (0.38-0.78)

0.58 (0.39-0.79)

0.67 (0.41-0.88)

0.70 (0.51-0.89)

0.77 (0.60-0.94)

TMTV

0.63 (0.43-0.83)

0.61 (0.40-0.81)

0.61 (0.41-0.82)

0.64 (0.40-0.87)

0.56 (0.36-0.77)

0.63 (0.36-0.91)

Abbreviations: progression-free survival (PFS), overall survival (OS), total metabolic tumor volume (TMTV), derived neutrophil-to-lymphocyte ratio (dNLR).

Second, we have calculated the likelihood ratio test (LRT) to measure the added prognostic value of TMTV and dNLR using log likelihoods of multivariable prognostic models without TMTV or dNLR and with TMTV plus dNLR (chi-square test: χ2). Results are provided in the Table S2.

CHEMO-IMMUNOTHERAPY GROUP

PFS

OS

Cox model variables

LR χ2

LR p value

LR χ2

LR p value

Multivariable model (with TMTV and dNLR)

6.9

-

12.9

-

Multivariable model without TMTV

3.5

0.06

9.0

0.04

Multivariable model without dNLR

3.8

0.08

8.3

0.03

Abbreviations: likelihood ratio (LR), progression-free survival (PFS), overall survival (OS), total metabolic tumor volume (TMTV), derived neutrophil-to-lymphocyte ratio (dNLR).

We have thus added a sentence in the Methods section/Statistical analysis: “The likelihood ratio test (LRT) for added prognostic value of TMTV and dNLR was obtained by comparing log likelihoods of multivariable prognostic models without TMTV or dNLR and with TMTV plus dNLR (chi-square test: χ2).

We also made changes with the addition of two sentences in the Results section:
OS: TMTV and dNLR added significant prognostic values to the multivariable model obtained with the backward elimination process for OS in the CIT group (p<0.05 for both; Table S4).”
PFS: “TMTV and dNLR didn’t provide any additional prognostic value to the multivariable model obtained with backward elimination for PFS in the CIT group (p>0.05 for both; Table S4)..

Comment #4: The test for median OS in high, intermediate and low risk in the CIT cohort was significant. Please state more clearly that the test is high vs. low - i doubt that there is a significant difference between 13.9 and 14.1 months.

This is an excellent suggestion. We have stated more clearly that the test is high-risk versus intermediate/low-risk groups in the Text and in the Table 4. Thank you for having highlighting this important point.

Round 2

Reviewer 1 Report

Even though the authors have made a significant effort to improve the presentation of the results of their work, I do not consider the findings of enough scientific relevance for publication.

Reviewer 2 Report

Thank you for your fine comments, I have nothing further.